# Obesity Associated with Prediabetes Increases the Risk of Breast Cancer Development and Progression—A Study on an Obese Rat Model with Impaired Glucose Tolerance

**DOI:** 10.3390/ijms241411441

**Published:** 2023-07-14

**Authors:** Prathap Reddy Kallamadi, Deepshika Esari, Utkarsh Reddy Addi, Rushendhiran Kesavan, Uday Kumar Putcha, Siddavaram Nagini, Geereddy Bhanuprakash Reddy

**Affiliations:** 1ICMR-National Institute of Nutrition, Hyderabad 500007, India; reddy.kp@icmr.gov.in (P.R.K.);; 2UT Southwestern Medical Center, Children Research Institute, Dallas, TX 75390, USA; 3Department of Biochemistry and Biotechnology, Annamalai University, Annamalinagar 608002, India

**Keywords:** breast cancer, obesity, metabolic syndrome, obese rat model, DMBA, insulin signaling, PI3K/Akt

## Abstract

Patients with comorbidities of obesity and diabetes are recognized to be at high risk of breast cancer development and face worse breast cancer outcomes. Though several reports showed the reinforced link between obesity, diabetes, and prediabetes with breast cancer, the underlying molecular mechanisms are still unknown. The present study aimed to investigate the underlying molecular link between increased risks of breast cancer due to coincident diabetes or obesity using a spontaneous obese rat model with impaired glucose tolerance (WNIN/GR-Ob rat). A single dose of solubilized DMBA suspension (40 mg/kg body weight) was orally administered to the animals at the age of 60 days to induce breast tumors. The tumor incidence, latency period, tumor frequency, and tumor volume were measured. Histology, immunohistochemistry, and immunoblotting were performed to evaluate the tumor morphology and expression levels of signal molecules. The development of mammary tumors in GR-Ob rats was characterized by early onset and shorter latency periods compared to control lean rats. While 62% of obese rats developed breast tumors, tumor development in lean rats was only 21%. Overexpression of ER, PR, Ki67, and p53 markers was observed in tumor tissues of obese rats in comparison with lean rats. The levels of the hallmarks of cell proliferation and angiogenesis involved in IGF-1/PI3K/Akt/GSK3β/β-catenin signaling pathway molecules were upregulated in obese rat breast tumors compared to lean rats. Furthermore, obesity with prediabetes is associated with changes in IGF-1 signaling and acts on PI3K/Akt/GSK3β/β-catenin signaling, which results in rapid cell proliferation and development of breast tumors in obese rats than the lean rats. These results indicate that tumor onset and development were faster in spontaneous obese rat models with impaired glucose tolerance than in their lean counterparts.

## 1. Introduction

Carcinoma of the breast is currently the leading cause of global cancer incidence with an estimated 2.3 million new cases in 2020, and the fifth leading cause of cancer mortality. Breast cancer is the most commonly diagnosed cancer among women accounting for 1 in 4 cancer cases and 1 in 6 cancer deaths [1]. Further, the incidence of breast cancer continues to increase. In the year 2023, 0.3 million new cases of breast cancer in the USA alone were reported [2]. Epidemiological investigations have established a strong association of breast cancer with metabolic dysregulation that leads to diabetes and obesity [3,4,5].

Obesity, a chronic disease with increasing prevalence both in wealthy nations as well as in low- and middle-income countries, has emerged as a global epidemic. Globally, the incidence of obesity has tripled since 1975 due to sedentary lifestyles and unhealthy dietary patterns. According to the WHO Report [6], over 1 billion people worldwide are obese which includes 650 million adults, 340 million adolescents, and 39 million children. It is estimated that 1 in 5 women and 1 in 7 men will be living with obesity by 2030 [7]. Likewise, there is a significant global prevalence of diabetes mellitus among 20–79-year-olds with an estimated 10.5% (536.6 million) living with the disease in 2021 which is predicted to rise to 12.2% (783 million) by 2045 [8]. Global estimates reveal that 1 in 10 people in the world are diabetic, while 3 in 4 diabetics live in low- and middle-income countries [8]. Most importantly, over 541 million adults have impaired glucose tolerance (IGT) referred to as prediabetes, which places them at high risk of type 2 diabetes (T2D). Furthermore, the probability of prediabetics developing diabetes is 3–10 folds higher than in normoglycemic individuals [9].

Accumulating evidence indicates a strong association between obesity, T2D, and several types of cancer, including breast cancer [10,11,12,13]. High body mass index (BMI), insulin resistance, increased levels of leptin and aromatase enzymes, and inflammation of breast adipose tissue are believed to contribute to obesity-related post-menopausal breast cancer [13]. In a recent study, high BMI, a widely used index of obesity, was an independent factor associated with a high 21-gene recurrence score in estrogen receptor (ER)–positive, ERBB2-negative young (≤45 years) breast cancer patients [14]. Several studies have provided compelling evidence to demonstrate an increased risk of breast cancer among diabetic women as well as higher mortality and diminished quality of life [15,16,17,18]. Diabetes as well as prediabetes is associated with the risk of breast cancer, especially in hormone-receptor-positive molecular subtypes [19].

Coincident obesity and T2D are recognized to increase the incidence of all molecular subtypes of breast cancer and worse outcomes, besides significantly lowering the survival rate [20,21,22,23,24]. Hyperglycemia, hyperinsulinemia, and insulin resistance, cardinal features of obesity and T2D, are believed to promote breast carcinogenesis [24,25]. A bi-directional relationship was observed between dysregulated glucose/insulin metabolisms with breast cancer [26]. In this clinical study, investigators found a correlation between severities of glucose/insulin metabolism with tumor and insulin resistance-related markers. The PI3K/Akt signaling pathway that plays a central role in various physiological processes and mediates the biological effects of insulin and insulin-like growth factor-1 (IGF-1) is aberrantly activated in diabetes/IGT, obesity, and breast cancer suggesting a possible link between these comorbidities [27].

The National Institute of Nutrition has developed a unique spontaneous mutant obese rat model with impaired glucose tolerance (WNIN/GR-Ob) that can be transformed into frank diabetes by dietary manipulations [28]. WNIN/GR-Ob rat displays a set of characteristics and features associated with the other obese animal models and in addition, exhibits impaired glucose tolerance (IGT), all of which make WNIN/GR-Ob rat a suitable model of metabolic syndrome (MetS) [29]. The present study was undertaken to investigate the combined effect of obesity and IGT on the development of chemically induced mammary tumors in the WNIN/GR-Ob rat model. Hormone receptor status, as well as the expression of Ki-67, p53, and key molecules in the IGF-1/PI3K/GSK3β/β-catenin signaling pathway, were assessed by immunohistochemical and immunoblotting analyses to evaluate the efficacy of 7,12-dimethylbenz[a]anthracene-(DMBA) induced mammary carcinogenesis in WNIN/GR-Ob rats as a coincident model of IGT/obesity and breast tumor.

## 2. Results

### 2.1. Obesity Accelerates the Onset and Development of Breast Cancer

The development of breast tumors was examined physically by palpation. No mammary tumors were detected in both control lean and control obese rats (Figure 1A,B). The onset of breast tumors occurred earlier in WNIN/GR-Ob rats administered with DMBA than in the lean rats with DMBA. The onset of tumor development in obese rats was observed after the 9th week of DMBA administration, whereas it was observed after the 26th week in counter lean rats. After 32 weeks of administration of DMBA, 62% of obese rats developed mammary tumors, while only 21% of the lean animals developed breast tumors (Figure 1C). The average latency period for tumor development was 119 days in obese rats administered with DMBA compared to 211 days in the lean rats administered with DMBA (Table 1). The average tumor latency was shorter in the obese rats by 92 days when compared to lean rats. It was clear that the onset of the tumor and its development was faster in obese rats than their counter lean rats with a higher percentage of tumor incidences in obese rats than the lean rats. Mammary tissues of all experimental rats were examined for histomorphological changes by H&E staining (Figure 2). Tumor mass showed lobules with pleomorphic epithelial cells with distinct variations in cell size and shape. Nuclear pleomorphism and increased mitotic figures were observed (Figure 2). Histopathological examination of these tumors revealed that in the obese rats administered with DMBA, 20% of the rats had adenocarcinoma and 40% had fibroadenoma, while all the tumors in lean rats were adenocarcinomas.

### 2.2. Obesity Promotes Oncogenic Markers Expression in Rat Mammary Gland

In the process of diagnosing breast cancer, the prognostic markers ER, PR, Ki67, and p53 proved to be the most effective. These proteins were screened for presence and relative expression in normal mammary tissues and DMBA-induced breast tumors of both lean and obese rats. Positive staining areas were observed for ER, PR, Ki67, and p53 in each and every breast tumor segment, and this was true regardless of whether the rats were lean or obese. The expression of these markers was significantly higher in rat mammary tumor tissues compared to normal mammary tissue. The expression of these molecules was also significantly higher in mammary tumor tissues of obese rats compared to lean rats (Figure 3).

### 2.3. Immunohistochemistry of Insulin Signaling Pathway and PI3K/GSK3β/β-Catenin Role in Breast Cancer Development

IGF-1 and its cognate receptor are crucial for the normal growth and development of the mammary gland. The overexpression of these molecules triggers signaling processes that are key to cancer cell growth and survival. IGF-1-mediated activation of PI3K/GSK/-catenin was investigated in order to study potential molecular processes that may contribute to the effect of obesity and impaired glucose tolerance on breast cancer development and progression. The concentrations of the IGF-1/PI3K/GSK3/β-catenin signaling pathway components IGF-1, IGF-1R, pIRS-1, PI3 kinase, Akt, pAkt, GSK-3, pGSK-3, β-catenin, and VEGF were evaluated. It was observed that the expression of signaling molecules IGF-1, IGF-1R, pIRS-1, PI3 kinase, pAkt, pGSK-3, β-catenin, and VEGF was elevated in the breast tumor tissues of obese rats compared to lean rats and that the expression of these signaling molecules was elevated in tumor tissue compared to respective control breast tissues (Figure 4 and Figure 5). The levels of total GSK-3 and Akt expression were reduced (Figure 4 and Figure 5). Further observations by immunoblotting also confirm that mammary tumor tissues of obese and lean rats had higher expression levels of PI3K, pAKT, pGSK-3β, β-catenin and VEGF than normal mammary tissues (Figure 6). The increased expression of IGF-1 and IGF-1R has stimulated the increased expression of downstream molecules, such as pIRS-1, PI3 kinase, pGSK-3, β-catenin, and VEGF, leading to the proliferation, survival, and metastasis of breast tissue.

## 3. Discussion

Breast cancer, the most frequent cancer in women, is a major public health issue. Several studies have provided strong evidence for a positive association between obesity, diabetes/IGT, and increased risk of breast cancer development as well as recurrent metastasis [30,31,32,33,34,35,36]. The combined effect of obesity and diabetes on breast cancer outcomes has been extensively reported in humans [21,22,23,37], and undiagnosed IGT is known to affect the survival of breast cancer patients [38]. Insulin/IGF signaling is reported to be dysregulated in obesity, diabetes, and breast cancer underscoring intricate overlaps in the underlying metabolic abnormalities and disease spectrum [39]. Hence, in this study, we investigated the combined effects of obesity and IGT on breast cancer risk and probable mechanisms using a genetically obese rat model with IGT for chemically induced breast cancers. The results of the present study reinforce the tenet that obesity and IGT have the propensity to progress to breast tumorigenesis.

In the present investigation, spontaneous mammary tumors did not develop in the untreated obese rats until 40 weeks of age. However, DMBA administration (40 mg/kg body weight) induced mammary tumors from the 9th week, which reached 62% at 40 weeks of age. DMBA administration accelerated the neoplastic transformation of the mammary gland in obese rats. On the contrary, the same amount of DMBA in lean littermates induced mammary tumors only at the 26th week, and only 21% of animals had tumors at 40 weeks. Obese rats developed mammary tumors faster after receiving the same dose of DMBA. Studies examined several different kinds of signs of mammary tumors and measured how much they were present in the breast tissue of both lean and obese rats that had been given DMBA to cause breast tumors. Acceleration of mammary tumors has been reported in mutant obese Yellow mice [40] and Zuccher rats [41,42] upon DMBA administration compared to counter leans supporting the current observations. However, these studies have not explained how breast cancer accelerates in these models. Previous investigations found that diet-induced or mutant-induced obesity increased tumor susceptibility. We found similar results with obese rats with decreased glucose tolerance in this investigation. This is the first report demonstrating enhanced susceptibility of a preclinical rat model with obesity and IGT to carcinogen-induced mammary tumor development. We believe that this can be a valuable animal model to analyze underlying molecular mechanisms associated with the development of comorbidities encompassing obesity, IGT, and mammary tumor and a valuable tool to test putative preventive/therapeutic agents.

ER and PR are important biological markers that have a key role in cellular growth, proliferation, and differentiation. Measurement of the levels of these hallmarks of breast cancer is useful as a prognostic indicator and in determining the possibility of hormonal resistance in breast cancer and treatment plan [43]. In the present study, obese animals had higher levels of all these markers in their tumor tissues than lean rats. Obesity has been suggested to increase steroid hormone receptor expression with consequent progression and proliferation of breast cancer cells [44]. An association between obesity and PR positivity was observed in ER-positive tumors [45]. The poorer survival of ER-positive breast cancer patients could depend on the tumor PR status [46]. Obese rats with IGT had a greater incidence of breast cancer than lean rats due to elevated ER and PR expression. IGT obesity may increase the risk of hormone-responsive breast cancer.

Ki67 is the most commonly used proliferative marker in breast cancer. High Ki67 expression predicts poor prognosis [47]. Ki67 distinguishes breast cancer molecular subgroups. It was classified as luminal-A or luminal-B based on the Ki67 value [48]. DMBA-induced breast cancers of obese rats had greater Ki67 expression with ER and PR positivity than lean rat breast tumors. Breast cancer has been clinically verified using the Ki67 as a proliferative marker [49]. The tumor suppressor gene TP53 encodes p53. In normal cells, ubiquitylation and proteasome activity destroy the p53 protein, which has a short half-life [50]. Mutations in the p53 gene stabilize a protein post-transcriptionally, causing cell accumulation [51], 18–25% of initial breast tumors have p53 mutations [52]. The IHC-detected p53 expression in breast cancer was associated with an aggressive, metastatic phenotype and worse outcomes [53,54]. Obese rats develop breast cancer faster than lean rats due to DMBA’s aggressive tumor induction and increased p53 expression in breast tumor tissues.

The major contributing factors of obesity and T2D/IGT that influence the risk of cancer were increased levels of growth factors such as insulin, IGF-1, steroid and peptide hormones, and inflammatory markers [55]. Aberrant activation of IGF-1 signaling has been documented in breast cancer tissues [56,57,58]. IGF-1 binds to its cognate receptor to induce phosphorylation of IRS-1 and triggers a cascade of events that eventually results in breast cancer development, progression, and metastasis [59,60]. ER is known to enhance the expression and activation of IGF-1R [61,62]. Activation of IRS-1 has been reported in ER-positive breast cancer [63]. Furthermore, the crosstalk between ER and IRS-1 increases the risk of breast cancer [56]. Although some studies on obese rodent animal models showed the association of obesity with increased breast cancer [64,65], the mechanism underlying the association between obesity and breast cancer has not been delineated. Here we demonstrate that administration of DMBA induced increased expression of IGF-1, IGF-1R, pIRS, and ER in the tumor tissues of obese rats with subsequent activation of downstream molecules in the signaling pathway that could promote tumor development.

Phosphorylation of IRS-1 stimulates PI3K/Akt signaling that plays a pivotal role in cell proliferation, cell survival, migration, and differentiation [66]. Inappropriate activation of PI3K/Akt signaling has been reported in diverse malignancies including breast cancer [67,68,69]. Our findings indicate that in obese breast tumors, PI3K stimulates PDK1, which phosphorylates Akt kinase. Akt phosphorylates GSK3 at Ser9, inhibiting its activity and stabilizing and accumulating β-catenin, which induces cell proliferation (Figure 7). The aberrant expression of β-catenin was associated with adverse outcomes of breast cancer [70,71,72]. The lower expression of these signaling molecules in lean rat tumor tissues may not stimulate this signaling cascade, resulting in delayed breast tumor induction and low tumor incidence.

In conclusion, the WNIN/GR-Ob rats serve as an excellent model for studying the influence of obesity and impaired glucose tolerance on the progression of chronic diseases, particularly breast cancer. GR-Ob rats treated with DMBA exhibited a higher incidence of mammary tumors at an earlier stage compared to lean rats. Obesity and IGT contribute to enhanced IGF-1 response, which facilitates cancer development by inhibiting apoptosis and promoting cell proliferation. The activation of the PI3K/Akt/GSK-3 signaling pathway by IGF-1, along with the nuclear accumulation of β-catenin, upregulates transcription factors associated with cell proliferation. Consequently, tumor development and progression were significantly elevated in obese rats, as evidenced by increased expression of VEGF and Ki67. These findings emphasize the importance of understanding the relationship between obesity and breast cancer in order to develop effective strategies for prevention and treatment.

## 4. Materials and Methods

### 4.1. Animal Grouping and Housing

Obese mutant rats with characteristics of abnormal response to glucose load (IGT), hyperinsulinemia, hypertriglyceridemia, hypercholesterolemia, and hyperleptinaemia were used in the study. Littermate-lean rats were used as controls. A total of 32 female rats (16 lean rats and 16 obese rats) 60 days of age were used for the investigation. The animals were housed individually in polycarbonate cages and autoclaved paddy husk was used as bedding material. Twelve hours of light–dark photoperiodicity with standard lighting conditions were maintained in the experimental rooms. The temperature, relative humidity, and air changes were kept constant at 22 ± 2 °C and 55 ± 10%, 14–16, respectively. Both the lean and obese rats were allowed 3 days of acclimatization and subsequently divided into four groups of eight animals each based on their body weight. Lean rats that did not receive any treatment served as lean control, whereas lean rats administered with DMBA were used as the lean tumor group. Untreated obese rats served as obese control, while obese rats that received DMBA administration are considered as the obese tumor group. All the animals received sterile standard rodent chow (AIN93M) diet and water ad libitum. The study was reviewed and approved by the Institutional Animal Ethical Committee (P10F/IAEC/NIN/5/2018/GBP/WNIN GR-Ob). The experiment was conducted in the animal facility, ICMR-National Institute of Nutrition, Hyderabad, India in compliance with the guidelines prescribed by the Committee for the Purpose of Control and Supervision on Experiments on Animals (CPCSEA).

### 4.2. DMBA Preparation and Administration

Mammary carcinogenesis was induced by the administration of DMBA (Sigma Aldrich, St. Louis, MO, USA). DMBA was dissolved in refined sesame oil (20 mg/mL) and stirred slowly using a magnetic stirrer until complete dissolution. A single dose of solubilized DMBA suspension (40 mg/kg body weight) was orally administered to the animals at the age of 60 days. The control rats received an equal amount of sesame oil. Based on a pilot study, since a single dose of DMBA at 40 mg/kg body weight was found to induce mammary tumors in both lean and obese rats without causing any mortality, this dose was administered for the experimental study.

### 4.3. Measurements of Tumor Growth Parameters

To determine the incidence and latency of tumor formation, rats were palpated at the thoracic and abdominal-inguinal mammary glands once a week starting from one week after the administration of DMBA. Tumor parameters such as the percentage of tumor-bearing animals per group (tumor incidence), the period from carcinogen administration to the appearance of the first tumor (latency period), the average tumor number per group (tumor frequency), and tumor size were measured. Tumor size was measured by recording the length and width of each tumor using a digital caliper in each group. The volume (V) of tumors was calculated according to the formula: V = π × S_1_^2^ × S_2_/12, where S_1_ and S_2_ are tumor diameters, assuming S_1_ < S_2_ [73].

### 4.4. Gross Necropsy

After completion of the feeding schedule (32 weeks after administration of DMBA), rats were fasted overnight and euthanized by CO_2_ inhalation. A gross necropsy of the rats was carried out to examine abnormalities. The mammary region of both tumor-bearing and control rats was shaved and the hair removed. The mammary tumors and control mammary tissues were excised and used for analysis. The in situ examination was carried out, after opening the viscera, and the major organs such as the brain, heart, lungs, liver, spleen, kidneys, and pancreas were separated from the viscera and cleaned from fat, blotted on a filter paper, and weighed (Sartorius analytical balance with a 0.1 gm sensitivity). The breast tumor tissues and normal breast tissues along with the other organs were fixed in a solution containing 10% formalin in sodium phosphate buffer at pH 7.4. These tissues were processed for histopathological and immunohistochemical analyses.

### 4.5. Hematoxylin and Eosin Staining

For analysis of tumor morphology, a smaller representative portion was taken from the freshly collected samples after animal necroscopy. For the histopathological study, H&E staining was performed for both breast tumors and normal breast tissues. Paraffin tissue sections of 4 μm thickness were made using a microtome (Jinhua Yidi Medical Appliance Co., Ltd., Jinhua City, China), followed by processing with an automatic tissue processor (Thermo Fisher Scientific Inc, Waltham, MA, USA). Hematoxylin and Eosin staining was performed using an autostainer (Sakura Tissue-Tek DRS 2000 automated slide stainer). The H&E stained slides were investigated for the detection of breast cancer metastatic stages, and images were acquired at 10× and 40× magnification with a Leica microscope. (Leica Microsystems, Wetzlar, Germany).

### 4.6. Immunohistochemistry

The formalin-fixed breast tissues were embedded in paraffin, and transverse sections (4 μm) were mounted in gelatin-coated slides. Immunohistochemical analysis was performed using a Vectastain Elite ABC kit (Vector Laboratories, Newark, CA, USA) that exploits the Avidin-Biotin Complex (ABC) method. Deparaffinized sections were processed in 10 mM sodium citrate buffer (pH 6.0) and heated for 5 min (antigen retrieval step). After blocking with 3% horse serum provided with the kit, the primary antibody (1:500 dilutions) was added to the sections and incubated overnight at 4 °C. In the present study, the following primary antibodies were utilized: estrogen receptor (ER, PA1-310B), progesterone receptor (PR, MA1-410), Ki-67 (MA5-14520) and vascular endothelial growth factor (VEGF, PA5-85171) from Invitrogen (Waltham, MA, USA), p53 (2524S), insulin-like growth factor 1 (IGF-1, 73034S), insulin-like growth factor 1receptor (IGF-1R, 3027S), phospho insulin receptor substrate 1 (pIRS-1, 3066S), phosphoinositide 3 kinase (PI3K, 4292S), protein kinase B (Akt, 9272S), p-Akt Ser473 (9271S), glycogen synthase kinase-3 beta (GSK-3β, 12456S), pGSK-3β Ser9 (9336S), and β-catenin (8814S) from Cell Signaling Technology (Danvers, MA, USA). After primary antibody incubation, the sections were washed three times for 5 min in 20 mM phosphate buffer saline (PBS pH 7.4). After washing with PBS, slides were incubated for 1 h at room temperature with a biotinylated secondary antibody (1:500 dilutions) solution and DAB was used as the chromogen. The negative controls were run simultaneously with the omission of the primary antibody. After staining, the sections were counterstained with hematoxylin. The sections were then dehydrated through ethanol and xylene before coverslips with Paramount. The DAB staining was visualized in the bright field using a Leica microscope (Leica microsystems, Wetzlar, Germany) at 10× and 40× magnifications.

### 4.7. Immunoblotting

The total proteins were extracted from breast tissues by homogenizing with 100 mM Tris-HCl buffer, pH 7.4 on ice with mortar and pestle, and the homogenate was centrifuged at 12,000× *g* for 30 min. The protein concentrations were estimated by the method of Lowry et al. [74] Equal amounts of protein from tissue extracts were separated using 12% SDS-PAGE and transferred to 0.2 µm nitrocellulose blotting membrane (GE Healthcare Life Science, Chicago, IL, USA) at 80 V for 1.5 h using a Bio-Rad transblot apparatus. To determine the uniformity of loading and transfer, membranes were stained with Ponceau S. The membrane was blocked for 2 h in phosphate buffer saline containing 20 mM sodium phosphate buffer, pH 7.4, and 5% nonfat dry milk powder at room temperature. Immunoblotting was performed by incubating the blot at 4 °C overnight with primary antibodies of IGF-1, phospho-IRS-1, PI3K, Akt, phospho-Akt, GSK-3β, phospho-GSK-3β, and β-catenin (1:1000 dilution). After overnight primary antibody incubation, the membrane was washed 3 times with PBS and then the blot was incubated for 2 h with HRP-tagged anti-rabbit/anti-mouse respective secondary antibody with a dilution of 1:10,000. Equal loading of the protein samples was assessed by probing the membrane with a 1:1000 dilution of the β-tubulin loading control antibody. The immunoblots were developed with enhanced chemiluminescence detection reagents (Bio-Rad Laboratories, Hercules, CA, USA). The images were analyzed and quantified using Image J software for Windows [75].

### 4.8. Statistical Analysis

All statistical analysis was performed using GraphPad Prism software 8.0 version. The data are expressed as mean ± standard mean error (SEM). *p* values were determined using one-way ANOVA followed by Tukey’s multiple comparison tests.

## Figures and Tables

**Figure 1 ijms-24-11441-f001:**
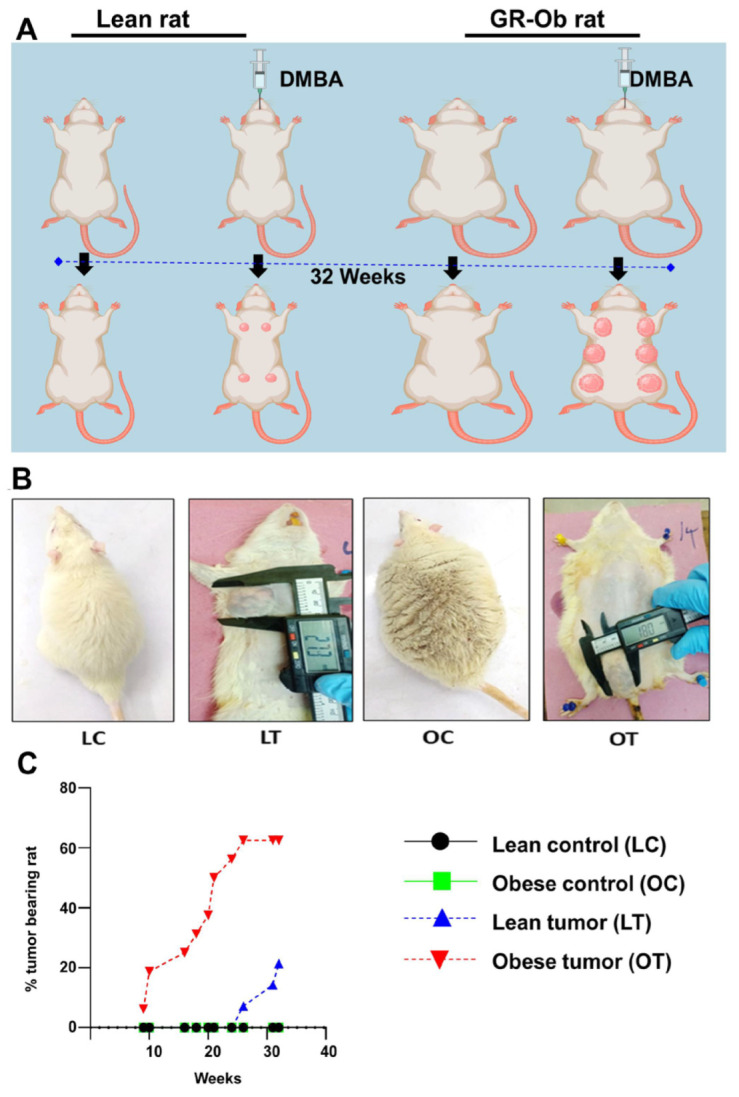
Induction and development of tumors in lean and obese rats upon DMBA oral administration (40 mg/kg body weight). (**A**) Graphical representation of tumor induction and development in lean and obese rats with DMBA administration. (**B**) Actual images of lean and obese rats with and without tumors. (**C**) Quantitative representation of the percentage of tumor-bearing animals with duration.

**Figure 2 ijms-24-11441-f002:**
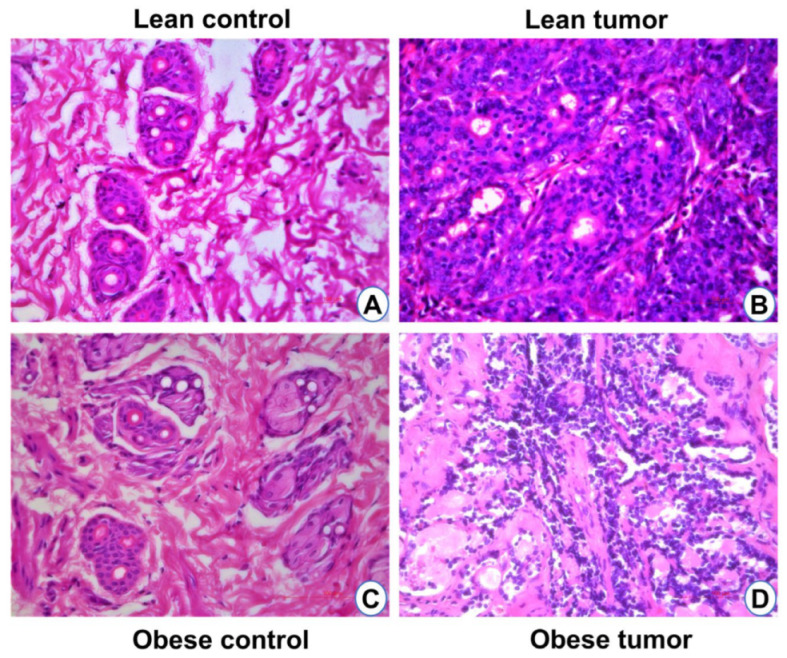
Histology (H&E staining) of breast tissues of lean control (**A**), lean tumor (**B**), obese control (**C**), and obese tumor (**D**). Tumor mass from both lean and obese rat breast tissues show lobules with pleomorphic epithelial cells and distinct variations in cell size and shape. Scale: 100 µm.

**Figure 3 ijms-24-11441-f003:**
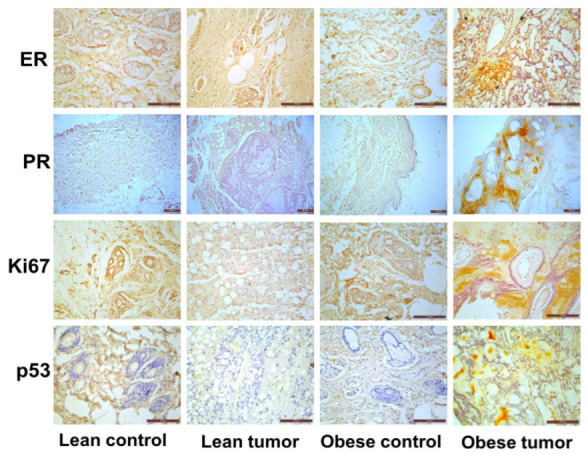
Expression of oncogenic markers in rat mammary gland. Representative images of immunohistochemical staining of ER, PR, Ki67, and p53 in breast tissues of lean control, lean tumor, obese control, and obese tumor. Scale: 100 µm (ER, Ki67 and p53) and 200 µm (PR).

**Figure 4 ijms-24-11441-f004:**
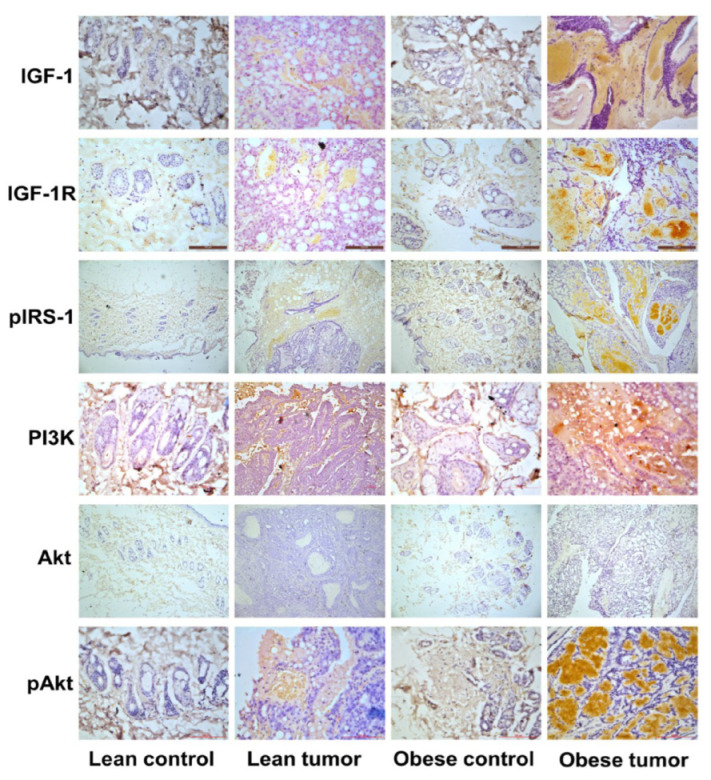
Expression of insulin and PI3K/Akt signaling pathway molecules by immunostaining. Representative images of immunohistochemical staining of IGF-1, IGF-1R, pIRS, PI3K, Akt, and pAkt in breast tissues of lean control, lean tumor, obese control, and obese tumor. Scale: 100 µm (IGF-1, IGF-1R, PI3K and pAkt) and 200 µm (pIRS-1 and Akt).

**Figure 5 ijms-24-11441-f005:**
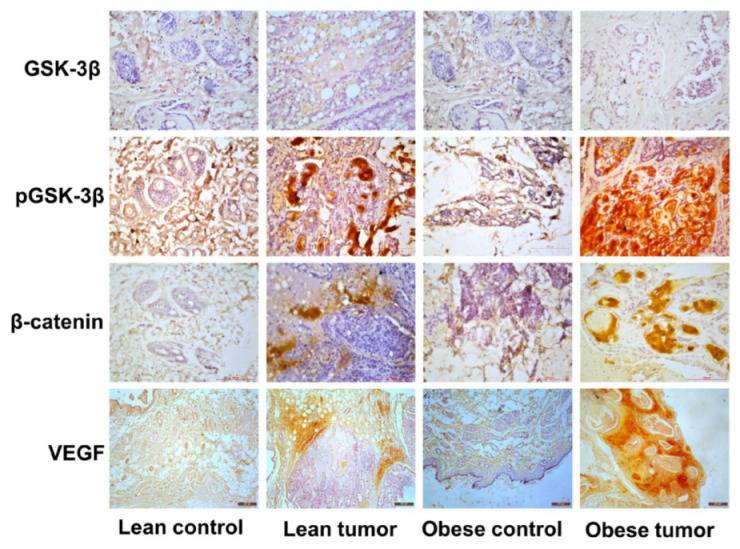
Expression of GSK/catenin signaling pathway molecules by immunostatining. Representative images of immunohistochemical staining of GSK-3β, p GSK-3β, β-catenin, and VEGF in breast tissues of lean control, lean tumor, obese control, and obese tumor. Scale: 100 µm (GSK-3β, pGSK-3β and β-catenin) and 200 µm (VEGF).

**Figure 6 ijms-24-11441-f006:**
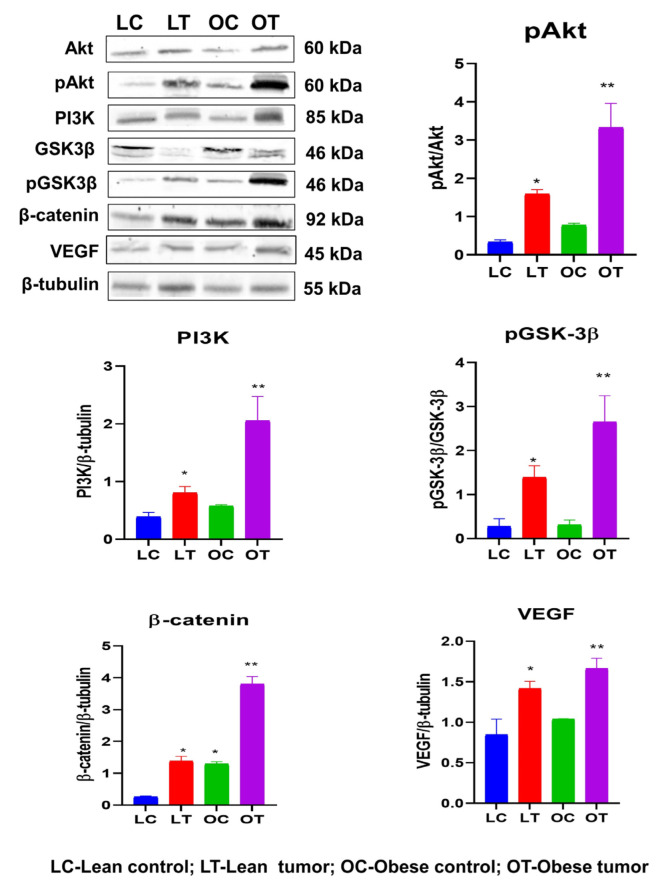
Expression of PI3K/Akt/GSK-3β/β-catenin signaling molecules by immunoblotting. Representative images of immunoblots along with respective quantitative bar graphs of pAkt, PI3K, pGSK-3β (ser9), β-catenin, and VEGF are shown. Data represent the ratio of phospho-form with total (pAkt and p GSK-3β) or ratio with β-tubulin (PI3K, β-catenin, and VEGF). Values are expressed as mean ± SEM of three replicates and statistical significance among the groups is indicated by * (*p* < 0.05) and ** (*p* < 0.01).

**Figure 7 ijms-24-11441-f007:**
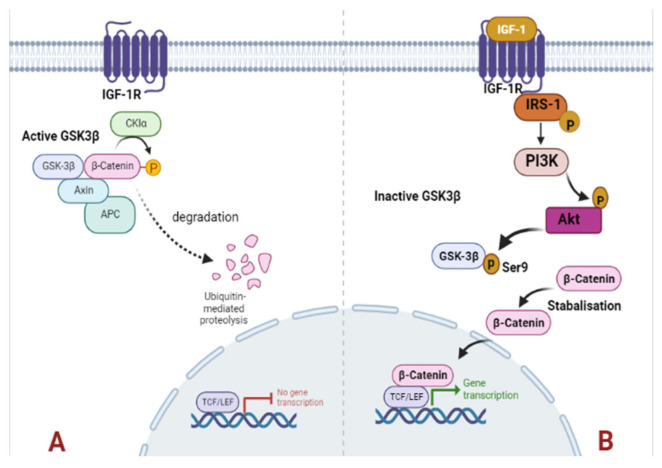
Schematic representation of IGF-1/PI3K/Akt/GSK-3β/β-catenin signaling. In obese rat breast tumor tissues, the higher-level expression of IGF-1 results in binding to IGF-1R. After binding of IGF-1 ligand to its receptor IGF-1R, IRS-1 is phosphorylated to initiate the downstream substrates PI3K. Subsequently, Akt is activated in response to PI3K signaling and becomes phosphorylated. The activated Akt phosphorylates GSK-3β at ser9, leading to GSK-3β inhibition and ultimately resulting in stabilization and accumulation of β-catenin which results in higher cell proliferation (**B**). In lean rats’ lower levels of IGF-1 cannot trigger the downstream cascade and reduced levels of IGF-1R, pIRS, pAkt, and GSK-3β (ser9) in lean tumor tissues subsequently, activated GSK-3β stimulates degradation of β-catenin (**A**).

**Table 1 ijms-24-11441-t001:** Tumor percentage, latency period average tumor volume, and the total number of tumors in obese and lean rats. Data are the mean of eight animals in each group.

	Control		DMBA-Treated	
	Lean	Obese	Lean	Obese
Incidence percentage	0	0	21.42	62.5
Latency period	0	0	211	119
Average tumor volume (cm^2^)	0	0	5.5	3.06
Cumulative tumor volume (cm^2^)	0	0	22.22	42.84
Total Number of tumors	0	0	4	14

## Data Availability

The data presented in this study are available with the corresponding author upon request.

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
