# Peer review of "Obesity Associated with Prediabetes Increases the Risk of Breast Cancer Development and Progression—A Study on an Obese Rat Model with Impaired Glucose Tolerance"

_ijms, 2023, doi:10.3390/ijms241411441_

Round 1

Reviewer 1 Report

Thank you for the opportunity to review the interesting manuscript by Kallamadi et al. 

1) Language should be improved in the text; scientific language is missing at many places. Please improve.

2) Lines 36-39, latest statistics data for the year 2022 is available and they can be used instead of 2021 stats.

3) Fig-2 The resolution of the images could be improved. Please check the histo figures- they seem to be 2D stretched.  If so please adjust in 3D.

4) Same with Fig-3. seems to have increased the image size in 2D. Also please remove some background letters or other characters in Fig3, 4. 

5) Please provide proper catalogue numbers of all the antibodies used. 

6) Fig6: In the Western blot, the authors should also show the totals or different phosphoproteins analyzed. Also, mention the molecular weight of the different proteins in the blots.

The language should be improved. Scientific language is missing at many places. 

Reviewer 2 Report

The study used a spontaneous obese rat model with impaired glucose tolerance to investigate the underlying molecular link between increased risks of breast cancer due to coincident diabetes or obesity. The potential significance is that it indentified IGF-1-mediated activation of PI3K/Akt/GSK-signalling with β-catenin nuclear accumulation as the underlying molecular pathway involved. However, there are some weaknesses. Firstly, it lacks glucose intolerance or insulin resistance data to confirm the mouse model. Secondly,  there is only the immunochemistry of the prognostic markers ER, PR, Ki67, and 123 p53,   western blot of these markes should also included to be more convincing. 

The quality of English is good. 
